# The Microbial Quality and Safety of Blenderised Enteral Nutrition Formula: A Systematic Review

**DOI:** 10.3390/ijerph17249563

**Published:** 2020-12-21

**Authors:** Omorogieva Ojo, Amanda Rodrigues Amorim Adegboye, Osarhumwese Osaretin Ojo, Xiaohua Wang, Joanne Brooke

**Affiliations:** 1Faculty of Education, Health and Human Sciences, School of Health Sciences, University of Greenwich, Avery Hill Campus, Avery Hill Road, London SE9 2UG, UK; 2Faculty of Health and Life Sciences, School of Nursing, Midwifery and Health, Coventry University Priory Street, Coventry CV1 5FB, UK; A.Adegboye@greenwich.ac.uk; 3South London and Maudsley NHS Foundation Trust, University Hospital, Lewisham High Street, London SE13 6LH, UK; Osarhumwese.Ojo@slam.nhs.uk; 4The School of Nursing, Soochow University, Suzhou 215006, China; wangxiaohua@suda.edu.cn; 5Faculty of Health, Education and Life Sciences, Ravensbury House, Birmingham City University, City South Campus, Birmingham B15 3TN, UK; joanne.brooke@bcu.ac.uk

**Keywords:** microbial quality, microbial safety, blenderised enteral nutrition formula, enteral nutrition formula, aerobic plate count, coliform

## Abstract

The use of blenderised enteral nutrition formula (ENF) is on the increase globally. However, concerns remain regarding the microbial quality and safety of blenderised ENF compared with standard recommendations and commercial ENF. Aim: This was a systematic review which sought to compare the microbial quality of blenderised ENF and commercial ENF and to evaluate the effect of storage time on blenderised ENF. Method: Four databases (Pubmed, EMBASE, PSYCInfo and Google scholar) were searched for relevant articles based on the Population, Intervention, Comparator, Outcomes framework. Results: Eleven studies which met the criteria were included in the systematic review. Two major areas were identified; Microbial Quality of Blenderised ENF versus Commercial ENF; and The Effect of Storage Time on Microbial Quality of Blenderised ENF. Overall, 72.7% of the studies showed microbial contamination in blenderised ENF compared with 57.1% of commercial ENF, and the storage time was another important factor in the rates of contamination. The extent of handling or manipulation of the enteral formula was critical in determining the level of contamination. Conclusion: Preparation techniques for blenderised ENF need to be established and caregivers taught how to prepare and administer it appropriately in order to reduce contamination. Further, well-designed studies are required, which compare the microbial quality of blenderised ENF using adequate handling techniques and commercial ENF.

## 1. Introduction

Enteral feeds are used in different clinical settings to support people who are malnourished or unable to maintain their own nutrition, and are usually in the form of pre-packaged, ready-to-use liquid feeds in most high-income countries [1,2,3]. These feeds are mostly sterile or microbial free preparations and nutritionally complete with energy, protein, vitamins and trace elements [1,4]. In this regard, the National Institute for Health and Care Excellence (NICE) [5] in the UK recommended that wherever possible, pre-packaged, ready-to-use enteral feed should be used in place of feeds that require decanting, reconstitution, and/or dilution, such as powder enteral feed, to reduce the risk of microbial contamination and infection. However, the use of blenderised enteral nutrition formula (ENF), which is any whole food mixture or liquid that is blenderised to be provided through an enteral tube and does not consist of water, medicine, or standard commercial enteral feeds, is on the increase worldwide [6,7,8]. These blenderised ENFs are often made at home by blending whole food or meal into a liquid that is thin enough to be given through a feeding tube [9,10]. Blenderised ENF can be made exclusively of food or a combination of food and standard commercial formulas [9].

There has been an increased interest in the use of homemade blenderised ENF, which is due to the fact caregivers and patients report a higher tolerance to real food blends, such as reduced gagging, retching, vomiting, diarrhoea, constipation, and an overall improved maintenance of weight [11,12,13]. The benefits of blenderised ENF include the impact of the variety of fresh and whole foods on health, such as the effect of phytochemicals and microbial abundance and diversity that are promoted by these foods [11,14]. Whole food blends may be preferable to commercial formulas in selected patients and allow the clinician and patient the ability to uniquely individualise the enteral nutrition plan [15,16]. However, there are also concerns regarding the use of these formulas due to the higher microbial load and the increased risk of food-borne infections [17,18]. While sterile ready-to-use commercial ENF are often associated with reduced risk of contamination and infection, it has been suggested that sterile production of blenderised ENF will be difficult to achieve in the home setting [17]. Enteral feed is an excellent medium for the growth and proliferation of food-borne microorganisms [19,20]. Some of the common microorganisms found in the enteral feeding system include bacteria such as Staphylococcus aureus and Clostridium difficile, and fungi [21,22]. In addition, other food-borne pathogens including Salmonella enteritidis have been implicated in food-borne diarrhoea and have been associated with enteral tube feeding [20,23].

Evidence of coliforms and fungi at unacceptable levels in blenderised ENFs has been reported [24]. There are also concerns that microbial contamination of blenderised ENF may be higher than in commercial ENF due to issues related to contaminated ingredients and equipment, and problems with handling and storage of the formula [9,24]. While the critical limit for the total microbial count of enteral feed samples is 10^1^ CFU/g at the start of feed administration, it should be 10^3^ CFU/g at the end [20]. For example, the U.S. Food and Drug Administration (FDA) guidelines recommend that food products are not acceptable for consumption if the aerobic counts exceed 10^4^ CFU/g in a single sample; 10^3^ CFU/g in 3 or more samples; the coliform count >3 organisms/g; and if the food products are positive for Listeria monocytogenes or Salmonella species [25,26]. However, studies have shown that most blenderised ENFs have standard plate counts greater than 10^1^ CFU/g [20].

The implications of administering contaminated enteral feed and the associated infections can be profound on patients. For example, patients who acquire an infection may develop anxiety, discomfort, inconvenience, delayed recovery, loss of confidence in the healthcare system, increased morbidity and mortality [19,27]. However, there have also been reports of the positive role of some microbes in gastrointestinal health, improved gastrointestinal function and reduced enteral tube feeding intolerance in patients fed blendersied ENF [17,24,28]. Accordingly, the enteral formula industry has responded with many commercially-prepared whole food blends that are available in a ready to hang form [11], although there are reported risks of food-borne diseases associated with homemade blenderised formulas [17,24,29]. Therefore, the current review is a systematic review that aims to evaluate the microbial quality and safety of blenderised enteral nutrition formula.

## 2. Methods

This review relied on the preferred reporting items for systematic reviews and meta-analyses (PRISMA) [30].

## 3. Types of Studies and Samples

The studies included were cross-sectional and in-vitro experimental studies. The samples analysed within all studies were blended ENF and commercial ENF.

## 4. Inclusion and Exclusion Criteria

The inclusion and exclusion criteria were based on the Population, Intervention, Comparator, Outcomes (PICO) framework [31] and are shown in Table 1.

## 5. Type of Intervention

Blenderised ENF, irrespective of the type of feeding tube and clinical setting, was the intervention of interest.

## 6. Types of Outcome Measures

### Primary Outcomes

Microbial counts of bacteria contamination in blenderised ENF and commercial ENF—aerobic bacteria count, coliform.

Microbial counts of bacteria contamination in blenderised ENF over time.

## 7. Search Strategy

Pubmed, EMBASE, PSYCInfo and Google scholar were the databases searched for articles of interest using keywords, Medical Subject Heading (MeSH)/synonyms and Boolean operators (AND/OR). The searches were conducted from the commencement of databases until 23 July 2020. Keywords were combined as follows; Enteral nutrition OR Blenderized formulas OR nutritional support OR home enteral nutrition OR enteral formula OR Enteral feeding OR blenderized enteral formula OR blended feeds OR Blenderized home-made food AND microbial contamination OR bacterial contamination. The strategy used for searching relied on the Population or Problem, Intervention, Comparator, Outcomes—PICO framework [31].

The process of selecting the studies included screening and evaluation for eligibility based on the PRISMA guidelines (Figure 1) [30]. This process of selection of articles involved two researchers (OO, OOO) who carried out the task independently, and the differences were resolved through discussion and consensus.

## 8. Data Extraction

All the articles found in all the databases were exported to EndNote (Analytics, Philadelphia, PA, USA) and the duplicates were removed. One researcher (OO) extracted the data from the studies and the other four researchers (OOO, X-HW; AARA; JB) cross-checked the extracted information. 

## 9. Evaluation of Quality

The quality of the cross-sectional study included was evaluated by one researcher (JB) using the Joanna Briggs Institute Critical Analysis Tool [32] and cross-checked by the other researchers (Appendix A). Only the information available in the study was used for quality evaluation.

## 10. Results

Three studies out of the eleven studies included were conducted in Iran [20,29,34] and two in the USA [25,28] (Table 2). In addition, two other studies were conducted in Brazil [36,38], and one study each in Costa Rica [33], the Philippines [37], Saudi Arabia [35] and the UK [17].

There were two distinct areas identified, namely: Microbial Quality of Blenderised ENF versus Commercial ENF; The Effect of Storage Time on Microbial Quality of Blenderised ENF. 

### 10.1. The Effect of Storage Time on Microbial Quality of Blenderised ENF

Johnson et al. [25] found that there was no S.aureus or coliform/E.coli detected at any time point following preparation. Furthermore, the total bacterial count was well below acceptable limits and all feeding formulas were acceptable for human consumption [25]. Reports from Madden et al. [17], Milton et al. [28] and Mahinkazemi et al. [29] were similar. While Madden et al. [17] showed that the impact of storage time on bacterial colony-forming units (CFU) varied with an increase in colonies on some agars but, overall, was not significantly different (feed A, *p* = 0.091; B, *p* = 0.764; C, *p* = 0.263), Milton et al. [28] observed that no sample had zero aerobic microbial counts although no substantial increase in microbial counts was observed during the 48 h. On the other hand, Mahinkazemi et al. [29] found that bacterial contamination (*S. aureus*, coliform) of blenderised tube feeding at the preparation time and 18 hrs after preparation did not change.

However, Baniardalan et al. [34] reported that the difference between the total viable contamination of the first and second sampling of blenderised ENF was significant (*p* = 0.004). In particular, after 18 h of preparation of the blenderised ENF, the coliform contamination increased by about 1.5 logs (*p* = 0.085) and *S. aureus* contamination also increased by about 2 logs (*p* = 0.008). In the Jalali et al. [20] study, there were significant increases in bacteria counts (coliform and *S. aureus*) 18 h after food preparation (*p*-value < 0.001).

Similarly, Mokhalalati et al. [35] demonstrated that there were significant increases over time in aerobic plate counts (APC) at each site (site 1, *p* = 0.023; site 2, *p* = 0.006; site 3, *p* = 0.042), and that for all blenderised tube feeding combined, there were significant increases in APC over time (*p* < 0.0005). Sullivan et al. [37] also found that the mean standard plate counts and mean coliform counts for blenderised samples taken immediately after tube feeding preparation increased significantly over time: (*p* = 0.008) and (*p* = 0.0005), respectively.

### 10.2. Microbial Quality of Blenderised ENF versus Commercial ENF

Table 3 shows the summary of results of the level of microbial contamination of blenderised ENF compared with commercial ENF and/or international standards. In the study by Mahinkazemi et al. [29], it was observed that the total coliforms of blenderised ENFs were less than 2 Most Probable Number (MPN)/g at both times. In contrast, six samples out of 18 (33%) commercial ENFs (powders) that were prepared on the wards were contaminated by coliform (6.41 ± 2.43 MPN/g), and E. coli was detected [29]. However, *S. aureus*, Salmonella and L. monocytogenes were not detected in either enteral formulas [23]. Baniardalan et al. [34] also found that the contamination of commercial ENF in all the samples was significantly (P < 0.05) more than that of the blenderised ENF. However, both the blenderised and commercial ENF were not contaminated with Salmonella spp. and L. monocytogenes.

In contrast, Sullivan et al. [37] noted that the natural food provided significantly higher mean standard plate counts than the commercial powder formula feeds at 1 and 2 h after preparation. This is supported by Vieira et al. [38], who also found that the counts of mesophilic and coliform bacteria were significantly higher in the non-commercial enteral diet and only 6.0% of the samples complied with the standard for coliform bacteria. Mokhalalati et al. [35] reported that the aerobic plate counts for all commercial ENF were not detectable (<10 CFU/gram) at all times. Additionally, while the maximum coliform count for any blenderised ENF sample from sites 1 and 2 was 50 CFU/gram, the coliform counts for all commercial ENF were not detectable (<10 CFU/gram) [35]. However, all the blenderised ENF and commercial ENF samples were negative for Salmonella and Staphylococcus aureus (<10 CFU/gram) [35].

In the study by Pinto et al. [36], the two types of enteral feeds showed contamination by coliforms and Pseudomonas spp. although there was no positive sample for Staphylococcus aureus and Salmonella spp. However, Listeria spp. was found in only one sample of handmade diets and the level of contamination was significantly higher in the handmade preparations (*p* < 0.05) [36]. Arias et al. [33] found that the degree of manipulation did not affect the microbial quality of blenderised ENF and that the level of contamination was as high as that in the commercial ENFs. Similarly, Johnson et al. [25] reported that the recipe selection and adherence to safe food handling of blenderised ENF provided safe feeding that was comparable to commercial ENF.

## 11. Discussion

The results of the studies included in this review showed varied outcomes in terms of the levels of bacterial contamination of blenderised ENF compared with commercial ENF. Similarly, the impact of storage time on bacterial colony-forming units of blenderised ENF differed between the studies. While some of the studies found that the increases in bacteria counts over time were not significant, the others demonstrated a significant increase.

However, eight of the eleven studies included, representing a proportion of 72.7%, reported that the level of microbial contamination of blenderised ENF was either a concern, higher than or not within the standard of international guidelines [17,20,33,34,35,36,37,38], compared to commercial ENF with four [29,33,34,36] out of seven studies (representing 57.1%). The differences between blendersied ENF and commercial ENF with respect to bacteria counts in this review may be due to a range of factors. For example, it has been suggested that microbial contamination in blenderised ENF may result from multiple phases including contaminated ingredients and equipment used for the blenderised ENF, poor hand hygiene, and storage and distribution processes that encourage microbial proliferation [24]. Borghi et al. [24] revealed there was 30–90% contamination in the open system of blenderised ENF and it was found to be associated with poor aseptic technique, poor cleaning and disinfection of equipment and ingredients that were contaminated. Madden et al. [17] also reported that the microbial loads of blenderised ENF are inevitable when the formulas are made from non-sterile ingredients and non-sterile conditions such as work surfaces, jugs, blenders and sieves, which are potential sources for microbial contamination. In the study by Mokhalalati et al. [35], almost all the blenderised ENFs had aerobic plate counts that were greater than 10,000 cfu/g compared with counts of less than 10 cfu/g (the detection limit) for all samples of commercial feeds.

According to Jalali et al. [20], 75% to 96% of blenderised tube feeding samples in the Philippines had standard plate counts that were greater than 10^1^ cfu/g, while higher contamination of blenderised ENFs was found in Saudi Arabia. Furthermore, almost all the samples of blenderised ENF studied in Saudi Arabia were reported to have aerobic plate counts greater than 10^4^ cfu/g [20]. Jalali et al. [20] noted this is particularly important as the Food and Drug Administration (FDA) has recommended that medical foods including enteral feeding formulas, containing more than 10^4^ cfu/g or if three or more samples exceed 10^3^ cfu/g, would require further action to be taken. 

Another area of interest is that some of the commercial ENFs examined in the studies in the current review included powder or commercial liquid formulas, which required different levels of handling, and these were compared with blenderised ENF. This element may explain why the results were not significantly different between the blenderised ENF and commercial ENF in some of the studies. According to Vieira et al. [38], the greater the need to handle enteral feeding formula, the higher the risk of microbial contamination. In particular, the level of microbial contamination decreases with the progression from blenderised ENF, which may be highly manipulated, to commercial powder formulas, and from commercial liquid formulas in cans to commercial formulas delivered in closed enteral feeding systems [35]. It has been reported that non-commercial and powdered commercial formula (which require decanting and reconstitution) show higher levels of microbial counts and lesser compliance with standards than the liquid commercial formula [38]. This is due to the procedure used for the non-commercial formulas (blenderised) such as inadequate cooking of contaminated raw foods and cross-contamination from food handlers and equipment, and the handling and hydration of the powder commercial formula, which are not required for the liquid commercial formula [38].

In terms of the effect of storage time on blenderised ENF, there is evidence that enteral feed, whether blenderised or commercial, provides an excellent medium for the growth of food-borne micro-organisms, which are of public health significance [20]. Furthermore, the contamination of enteral feed increases the risk of nosocomial infections including diarrhoea and pneumonia [20]. Therefore, strategies for reducing microbial contamination of enteral feeding formulas need to be developed to reduce the risk of microbial contamination. For blenderised ENF, preparation techniques need to be established and caregivers taught how to prepare and administer ENF appropriately to reduce contamination. In addition, cleaning and sanitizing of kitchen equipment and surfaces needs to be embedded in the preparation of blenderised ENF [28]. Appropriate blenders and equipment need to be cleaned thoroughly through disassembling and effective sanitisation, alongside the need to ensure adequate home environments including water source, refrigeration temperature and storage capacity for blenderised ENF [28]. Brown et al. [9], acknowledged that food safety is a concern in blenderised ENF; therefore, it is essential that the foods are cooked thoroughly, and are kept at the correct temperature. It has also been recommended that blenderised ENF should be administered in the form of bolus feed instead of continuous infusion [9]; this supports its safe administration, rather than allowing the blenderised ENF to be at room temperature for more than 2 h [9].

Furthermore, recommendations for the use of sterile (commercial) ENF have been made when there is a risk of aerobic counts of non-sterile feeds [37,39]. For example, NICE [5] recommended that wherever possible, pre-packaged, ready to use feeds should be used instead of feeds that would be decanted, reconstituted or diluted, such as the powdered commercial formulas. In addition, the system selected should allow for minimal handling to assemble and should be compatible with the patients’ enteral feeding tube [5]. Therefore, sterile, non-manipulated closed systems of enteral nutrition administration, good hygiene and maintaining microbial surveillance are the preferred approaches to reducing microbial contamination and managing enteral nutrition provision [40].

## 12. Limitations

Most of the blenderised ENF microbial studies were conducted in countries and conditions where the ambient temperatures are higher than those expected in hospitals and homes of developed countries. In addition, the methods of blenderised ENF preparation in those studies were not well described; therefore, it may be difficult to ascertain whether they were prepared properly. Many of the studies were conducted a few years ago and storage and refrigeration techniques could also have skewed the results.

## 13. Conclusions

This systematic review has demonstrated there are significantly higher levels of bacteria contamination in blenderised ENF compared with commercial formula. Furthermore, there are also differences in terms of the level of bacteria contamination between blenderised ENF and commercial formula based on storage time. Therefore, preparation techniques for blenderised ENF need to be established and caregivers taught how to prepare and administer it appropriately to reduce contamination. More well-designed studies comparing the microbial quality of blenderised ENF using adequate handling techniques and commercial ENF are required. Furthermore, it will be useful for future studies to explore the effect of poor handling of ENF on health outcomes.

## Figures and Tables

**Figure 1 ijerph-17-09563-f001:**
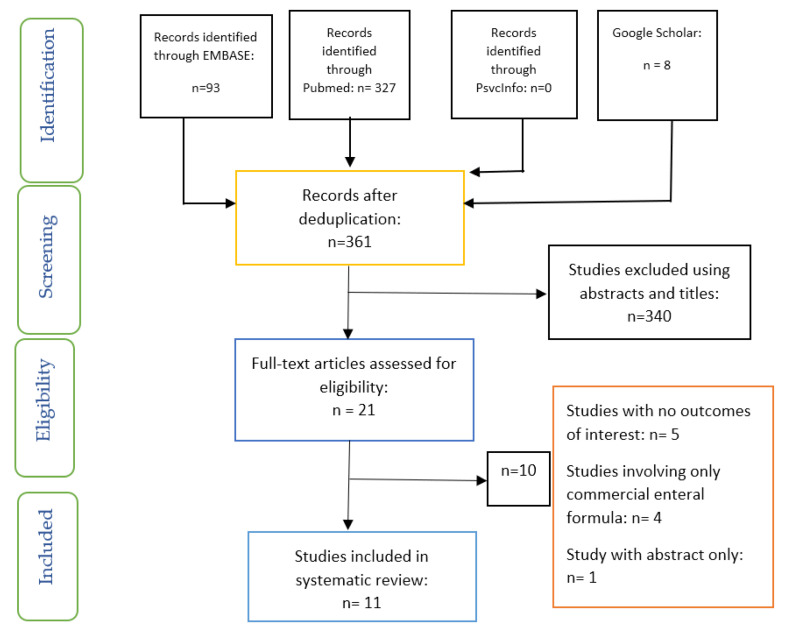
Preferred reporting items for systematic reviews and meta-analyses (PRISMA) flow chart on the selection and inclusion of studies.

**Table 1 ijerph-17-09563-t001:** Inclusion and exclusion criteria.

	Inclusion Criteria	Exclusion Criteria
**Population or Problem**	Patients (adults) on blended tube feeding or blenderised enteral nutrition formula	Studies involving children aged below 18 years
**Intervention**	Blended enteral nutrition formula at 0 h time point	Individuals on normal oral dietary intake
**Comparator**	Commercial enteral nutrition formula or Blended enteral nutrition formula at later h time point	Parenteral nutrition, parenteral plus enteral nutrition
**Outcomes**	Primary Outcomes: a. Microbial counts of bacteria contamination in blenderised ENF and commercial ENF; b. Microbial counts of bacteria contamination in blenderised ENF over time.	Qualitative outcomes such as patient feelings
**Types of Study**	Quantitative studies	Letters, comments, reviews, qualitative studies

Abbreviation: Enteral nutrition formula (ENF); hour (h).

**Table 2 ijerph-17-09563-t002:** Shows results of the data extracted from the studies included.

Citation	Country	Aim/Objective of Study	Study Design	Study Method/Sample Size/Description	Age (Years)	Study Results/Conclusion
Arias et al. [33]	Costa Rica	To assess the level of microbial contamination in enteral feeds in hospitals.	In-vitro experimental study	A total of 124 enteral feeding solutions were assessed. Overall, 50% of the samples were made from commercial formula (Ensure^®^) and the remaining 50% were solutions prepared at the nutritional hospital services.	Not Applicable	The level of gram-negative bacteria in the enteral feed samples varied from 10^3^ to 10^7^ CFU/mL. Enterobacter cloacae, Escherichia coli, Serratia sp. and Klebsiella pnuemoniae were the most frequently isolated coliforms. Pseudomonas sp. was isolated in more than 70% of the samples made from commercial solutions, fruits and vegetables.
Baniardalan et al. [34]	Iran	To evaluate and compare the bacteria safety of hand-made blenderised ENF and commercial ENF.	In-vitro experimental study	Seventy samples including 21 handmade formulas sampled at two sampling times (the time of preparation and 18 h after preparation, and 28 commercial ENF) were studied.	Not Applicable	The microbial safety of enteral feeding solutions in this hospital was found to be much lower than the standard values.
Jalali et al. [20]	Iran	To evaluate the microbial quality of blenderized ENF in two university hospitals.	In-vitro experimental study	A total of 152 samples (76 samples each at the time of preparation and 18 h following preparation) were collected. Standard plate count, coliform count and *Staphylococcus aureus* count in all samples were conducted. The presence of *Salmonella* spp. and *Listeria* spp. were also examined.	Not Applicable	It was found that most of the BTF in the hospitals were not safe. Compared to standard levels, the BTFs were found to be highly contaminated and this could be a source of significant risk in the development of food-borne disease or nosocomial infection.
Johnson et al. [25]	USA	To compare microbial levels of a standard commercial formula (CF), a BTF made using baby food (BTF-BF), and a BTF prepared from blending whole food (BTF-WF).	In-vitro experimental study	Three tube-feeding formulas (CF, BTF-BF, BTF-WF) were compared for the growth of aerobic microorganisms, *S. aureus*, coliforms, and E. coli, at zero hour, 2 h, and 4 h after tube feed preparation.	Not Applicable	It was found that BTF recipe selection and adherence to safe food handling provide a safe feeding that is comparable to CF in the hospital setting.
Madden et al. [17]	UK	To examine the risks of blended feed in providing nutritionally adequate intake.	In-vitro experimental study	The blended feed was made using three different methods (professional, jug and stick blenders) and three storage procedures. Feed samples were diluted and bacterial colony-forming units (CFU) were counted.	Not Applicable	The level of bacterial contamination was a concern. However, this was not due to the methods of preparation or storage used.
Milton et al. [28]	USA	To examine the procedure for minimising bacterial growth of BTF.	In-vitro experimental study	BTF was assessed for the growth of aerobic microorganisms including; Escherichia coli, Staphylococcus aureus, and coliforms at baseline, 24-h, and 48-h intervals after preparation for a total of 150 colony forming units (CFU) counts performed.	Not Applicable	It was concluded that safe food-handling procedures can reduce bacterial contamination of BTF and can also reduce the risk of food-borne infection in HEN patients.
Mahinkazemi et al. [29]	Iran	To examine bacterial contaminations of enteral feeding (EF).	In-vitro experimental study	A total of 54 EF samples; 36 blenderised tube feedings (BTFs) and 18 commercial powder feedings (CPFs) of patients in the intensive care units were examined.	Not Applicable	The issues of quality, safety, and the appropriate type of enteral nutrition formulas are essential based on the bacterial contamination of CPFs.
Mokhalalati et al. [35]	Saudi Arabia	To compare the microbial safety of BTF and commercially prepared formulas (CPF).	In-vitro experimental study	Eighteen samples of BTF were collected from 3 hospitals. Samples of a CPF were also collected for comparison.	Not Applicable	BTFs are highly contaminated and may increase the risk of nosocomial infections.
Pinto et al. [36]	Brazil	To assess the microbiological quality and aseptic conditions in the preparation and administration of handmade and commercial ENF.	In-vitro experimental study	Twenty-five samples of enteral diets were analysed, 13 of them were non-industrialized diets (prepared in the hospital facility) and 12 were industrialized diets, collected in two different times (immediately after the completion preparation (T0) and after administration to the patient (T1). There were 50 sample units, each of them containing at least100 mL of the diet.	Not Applicable	The microbial quality of the enteral feeds was not satisfactory. The aseptic conditions in the hospital concerning preparation and handling of enteral diets increases the risks of cross-contamination.
Sullivan et al. [37]	Philippines	To assess the microbial quality of BTF.	In-vitro experimental study	Two feedings were prepared on three separate days at four hospitals. The tube feedings were either blended foods or commercial products. Samples of each feeding were collected for coliform count and standard plate count at the time of preparation and different times after preparation.	Not Applicable	The microbial quality of most of the hospital-prepared enteral tube feedings was not within the published guidelines for microbial safety.
Vieira et al. [38]	Brazil	To evaluate the microbial quality of commercial ENF and blenderised whole foods ENF.	Cross-sectional study	A total of 66 samples of commercial (CD, n = 33) and noncommercial (NCD, n = 33) enteral diets were collected at the homes of patients on HEN.	73 years (20–100 years)	The homemade blenderised ENF contained high levels of bacterial contamination.

Abbreviations: Blenderised tube diets (BTD); blenderized tube feeding (BTF); Colony forming units (CFU); home enteral nutrition (HEN); home enteral tube feeding (HETF).

**Table 3 ijerph-17-09563-t003:** Summary of results of the level of microbial contamination of blenderised ENF compared with commercial ENF and/or international standards.

Citation	Level of Microbial Contamination of Blenderised ENF	Level of Microbial Contamination of Commercial ENF	Comments
Arias et al. [33]	↑	↑	The concentration of gram negative rods found in the samples of enteral feeding solutions ranged from 10^3^ to 10^7^ CFU/mL, significantly exceeding the permissible level (10^2^ CFU/mL or less). There were no differences in the levels of contamination of the formulas. Blenderised ENF made from fruits or cooked vegetables showed contamination levels as high as the ones present in the commercial based solutions (Ensure^®^).
Baniardalan et al. [34]	↑	↑	The contamination of commercial formulas in all three microbiological samples was significantly more than that for handmade samples. Overall, 76% of handmade samples had total viable counts greater than 10^3^ CFU/g compared to 96% of commercial formulas at the time of preparation.
Jalali et al. [20]	↑	Not Applicable	In the standard plate count, 97% of the samples had counts greater than 10^3^ CFU/g, while 71% had counts greater than 10^4^ CFU/g at the time of preparation.
Johnson et al. [25]	Total bacterial count was well below acceptable limits	Total bacterial count was well below acceptable limits	All 3 feeding formulas at zero hour, 2 h, and 4 h for each of the 3 sampling dates were acceptable for human consumption.
Madden et al. [17]	↑	Not Applicable	The bacterial load of *Enterobacteriaceae* of approximately half of the blended feeds was categorised as unsatisfactory (i.e., CFU/g > 10^4^), with no clear pattern of association with preparation or storage method.
Milton et al. [28]	At time of preparation and after 24 h, 10% had a CFU count of >10^4^, and, at 48 h, 12% exceeded 10^4^ CFUs.	Not Applicable	The result showed that 88% of the samples met the US Food Code criteria for safe food consumption; 10.7% met guidelines for marginal safety by other standards; and 1.3% slightly exceeded 10^5^ CFUs.
Mahinkazemi et al. [29]	Bacterial contamination (*S. aureus*, coliform) of blenderised ENF at the time of preparation and 18 h after preparation were <10^1^ CFU and <2 MPN/g, respectively.	Overall, 33% of commercial ENF which were prepared in the wards had coliform contamination of 6.41 ± 2.43 MPN/g and E. coli was detected.	The presence of E. coli and coliforms in 33.3% of commercial ENF showed that these were at an unacceptable level of contamination.
Mokhalalati et al. [35]	↑	↓	Overall, 86% of standard blenderised ENF and therapeutic blenderised ENF had Aerobic Plate Count (APC) >10^4^ CFU/g, while for all commercial ENFs, APC was not detectable (<10 CFU/g) at all times. The maximum coliform count for any blenderised ENF sample from sites 1 and 2 was 50 CFU/g. Coliform counts for all Commercial ENFs were non detectable (<10 CFU/g).
Pinto et al. [36]	↑	↑	Both kinds of blenderised ENF and commercial ENF showed contamination by coliforms and *Pseudomonas* spp. *Listeria* spp. was detected in only one sample of handmade diets. However, contamination was significantly higher in the blenderised ENF (*p* < 0.05) compared with Commercial ENF.
Sullivan et al. [37]	↑	Not Applicable	Overall, 38% of blenderised ENF had coliform counts greater than 10 MPN/g, and 92% of the samples had standard plate counts greater than 10^3^ CFU/g. There were significant increases in mean coliform and standard plate counts over 4 h.
Vieira et al. [38]	↑	Samples of powder commercial ENF complied less when compared to liquid commercial ENF.	Only 6% of samples of blenderised ENF met the standard for coliform.

Abbreviations/Symbols: Colony forming units (CFU); enteral nutrition formula (ENF); most probable number (MPN); high level of contamination (↑); low or non-detectable level of contamination (↓).

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
