# Peer review of "The Microbial Quality and Safety of Blenderised Enteral Nutrition Formula: A Systematic Review"

_ijerph, 2020, doi:10.3390/ijerph17249563_

Round 1

Reviewer 1 Report

Some of my concerns as the same as in the previous draft.

Title:  do you mean in adult populations?  We don't know if adults and children were provided the various formulas in the studies cited.  Since microbial cut points are not based on age, why restrict to adult populations only?

  1. Why is the Hurt study (ref 6) still included in the table and why is it used to report adverse events related to blenderized tube feeding (BTF)? It was a survey of 54 adults using blenderized tube feeding and they reported good outcomes. However, on page 8 of 14, the authors state they reported adverse events.  However, this is what the authors of that paper stated:   ï‚— 54 tube fed adults seen in the Mayo Clinic Rochester were surveyed.
    ï‚— BTF was used by 55.5% of patients.
    ï‚— 90% expressed a desire to use blenderized food tube feeding (BTF), if provided with adequate information.

    â—¦ 83% had no symptoms on BTF
    â—¦ 67% had no symptoms on commercial formula
    Results: BTF is tolerated as well as commercial formulasSince there is no assessment of microbial contamination and outcomes were reported as good, t
    he authors are misrepresenting the results of this article. It does not belong in the table.
  2. The Barniardalan (ref 34) article should note the fact that the tube feeding was left at room temperature for 18 hours!  No wonder they had high bacteria counts. (see bottom of page 8). Also, this should be noted- in that study, 96% commercial formulas were contaminated at baseline and 76% of BTF contaminated at baseline!  This is very important.
  3. The table summary needs the following information: Mokhalalati (ref 35) found 86% of BTF and 0% of commercial formula exceeded CFU standards at baseline. The interpretation of this is that the BTF was not properly prepared- it was contaminated in the preparation process.  The interpretation of the Mahinkazemi (ref 29) study is confusing. According to the authors, contamination was found in both the BTF and the reconstituted commercial powder formula. 
  4. The Milton study was carried out in a home enteral nutrition setting and found almost 90% met the US standards required of commercial formula. Only one sample exceeded 10(5) and NONE exceeded 10(6). These were families prepared BTF using expected safe food handling practices in the US in their homes.
  5. The Madden paper indicates PCR technique identified several bacterial species but the only counts reported were Bacillus cereus (which was in acceptable limit) and Enterobacteriaceae of approximately half the blended feeds was categorised as unsatisfactory, i.e. CFU/g >10(4) but for no apparent reason (possibly a particular recipe item was contaminated at baseline). 
  6. Jalili (ref 20) 98% of the BTF was contaminated at baseline (indicates unsafe food handling practice or a contaminated BTF ingredient).
  7.  Finally, Arias compared tube feeding substrates from 5 different hospitals in Costa Rica in 2003.  He found unacceptably high levels of contamination throughout the samples and importantly: " in this study no differences in the levels of contamination of the formulas were observed. Complex preparations such as those prepared with fruits or cooked vegetables showed contamination levels as high as the ones present in the commercial based solutions (Ensure® )"  Furthermore, he stated since contaminating microbes were subject to destruction by heat, the contamination had to be from improper handling/prep methods instead of the feeding substrate.

Overall, I feel that the paper is unjustifiably slanted against BTF use.  The studies over all show that if safe food handling is used, BTF can be safe.  It also shows that if safe food handling methods are not used, commercial formulas can be unsafe.  There is a growing body of work demonstrating the safety and efficacy of whole food based tube feeding due to its positive impact on the gut microbiome.  This review represents a wonderful opportunity to showcase the fact that BTF can be given and the benefits enjoyed IF safe food handling is employed. Patient environments and patient capacity to provide safe tube feeding need to be assessed and oversight by trained dietitians is vital for all tube feeding. It also should point out that just because commercial formula is used, it doesn't mean the patients are getting a safe feeding. 

Finally and most importantly, not one single study has been published showing a connection to bacterial/food borne-illness from patients on BTF.  If this has happened we certainly need to know about it, but none of the articles reviewed here report it. 

A survey of British dietitians found they all had concerns about bacterial infection of patients using BTF but none had ever encountered it in their patients. 

Author Response

Reviewer 1:

Comments: Some of my concerns as the same as in the previous draft.

Response: Thank you for your comments.

We apologise for not fully addressing some of your concerns. We have reviewed our responses to your comments in the first round of the review process and observed that we incorporated all your suggestions in the revised manuscript except for the Hurt et al study which we retained and provided a rationale for our decision.

However, we have now removed the Hurt et al study from the articles included in the review and reproduced our responses to your earlier comments in the first round of the review process;

 Comments: The authors have reached the conventional conclusion that blenderized tube feeding (BTF) is not sanitary though they agree the literature is mixed. Indeed, the overall p-value was 0.04.  They conclude that ready to hang commercial formulas are the most appropriate enteral feeding. However, perhaps the conclusion should be that BTF preparation techniques need to be established and caregivers taught how to prepare and administer it appropriately in order to reduce contamination. Reports of improved tolerance to BTF compared to commercial standard formulas are widely reported. It would not be ethical to deny tube fed patients the advantages of a BTF that is prepared correctly.

Response: Thanks for this relevant comment. The discussion and the conclusion have been revised and the following sentence has been included;

Therefore, preparation techniques for blenderised ENF need to be established and caregivers taught how to prepare and administer it appropriately in order to reduce contamination.

Comments: The authors should explain the increased interest and use of BTF is due to the fact that caregivers and patients report better tolerance to a real food blend (reduced gagging, retching, vomiting, diarrhea, constipation) and overall improved achievement of weight/growth goals. Line 88 on page 2 states "the potential benefits of blenderised ENF have not been well researched" but that is not true.  There are several convincing publications demonstrating the superiority of a whole food blend over commercial formula (see ref list below).

Response: Thanks for raising this issue and pointing out relevant literature. We agree that the introduction understated the benefits of blenderised ENT. We revised the introduction and included the potential benefits of blenderised ENT and cited some of the references recommended.

Comments: Nutrition experts tout the advantages of a whole food/plant based diet that confers intestinal microbiome diversity and improved health outcomes; in contrast, traditional commercial formula is highly processed and monotonous with no representation from plant sources. The message that a whole food/plant based diet is advantageous has reached the tube feeding population and they want to utilize this feeding substrate. Accordingly, the enteral formula industry has responded with many commercially-prepared whole food blends that are available in a ready to hang form (Bennett K, Hjelmgren B, Piazza J. Blenderized tube feeding: health outcomes and review of homemade and commercially prepared products. Nutr Clin Prac. 2020;35(3):417-431).  Furthermore, a BTF can be made by adding sterile baby food to a commercial formula- providing advantages of food and advantages of sterile products.

Response: Thanks for raising this issue and pointing out relevant literature. We agree that the introduction understated the benefits of blenderised ENT. We revised the introduction and included the potential benefits of blenderised ENT and cited some of the references recommended.

Comments: Note that prior to the work by Johnson and the work by Milton, BTF microbial studies were conducted in countries and conditions where the ambient temperatures are higher than expected in hospitals and homes of developed countries. Also, the methods of BTF preparation in those studies were not described so we cannot be sure that they were prepared properly. More importantly, no study has specifically linked BTF to increased food borne illness in tube fed patients. Another key feature of these studies is that subjects receiving commercial formula also had more oversight from healthcare providers because they had better insurance/medical provider programs.  Many of the patients using BTF did not.  Perhaps it is oversight and guidance on enteral feeding that makes the difference instead of the feeding substrate.

Response: The following sentence has been included as part of the limitations;

Most of the blenderised ENF microbial studies were conducted in countries and conditions where the ambient temperatures are higher than those expected in hospitals and homes of developed countries. In addition, the methods of blenderised ENF preparation in those studies were not well described, therefore, it may be difficult to ascertain whether they were prepared properly.

Comments: Finally, I am not sure why the 2015 Hurt study is included in the analysis. There was no investigation of bacterial load of BTF. This was a survey of their patients on home enteral nutrition- many using full or partial blended tube feeding.  No formula was collected and tested. The survey showed that participants experienced much better outcomes on BTF compared to standard commercial enteral formula (see abstract below).

Response: Thank you for your comments. The Hurt study was only included in the systematic review to provide a background to the adverse events and was not included in the initial meta-analysis. We have now decided to limit this work to only systematic review without meta-analysis.

Second Round of the Review Process:

Comments: Title:  do you mean in adult populations?  We don't know if adults and children were provided the various formulas in the studies cited.  Since microbial cut points are not based on age, why restrict to adult populations only?

Response: Thank you for your comment. We included the adult population based on the suggestion by an earlier reviewer. We have now deleted it from the title as we agree with your recommendation.

Comments:

  1. Why is the Hurt study (ref 6) still included in the table and why is it used to report adverse events related to blenderized tube feeding (BTF)? It was a survey of 54 adults using blenderized tube feeding and they reported good outcomes. However, on page 8 of 14, the authors state they reported adverse events.  However, this is what the authors of that paper stated:   Â— 54 tube fed adults seen in the Mayo Clinic Rochester were surveyed.
    — BTF was used by 55.5% of patients.
    — 90% expressed a desire to use blenderized food tube feeding (BTF), if provided with adequate information. 
    â—¦ 83% had no symptoms on BTF
    â—¦ 67% had no symptoms on commercial formula
    Results: BTF is tolerated as well as commercial formulasSince there is no assessment of microbial contamination and outcomes were reported as good, the authors are misrepresenting the results of this article. It does not belong in the table.

Response: The Hurt et al study has been removed from the studies included in the main review as we have now focused primarily on the microbial quality of blenderised ENF and the effect of storage time and not on adverse events. The Hurt et al study now provides background to the review.

Comments:

  1. The Barniardalan (ref 34) article should note the fact that the tube feeding was left at room temperature for 18 hours!  No wonder they had high bacteria counts. (see bottom of page 8). Also, this should be noted- in that study, 96% commercial formulas were contaminated at baseline and 76% of BTF contaminated at baseline!  This is very important.

  1. The table summary needs the following information: Mokhalalati (ref 35) found 86% of BTF and 0% of commercial formula exceeded CFU standards at baseline. The interpretation of this is that the BTF was not properly prepared- it was contaminated in the preparation process.  The interpretation of the Mahinkazemi (ref 29) study is confusing. According to the authors, contamination was found in both the BTF and the reconstituted commercial powder formula. 

  1. The Milton study was carried out in a home enteral nutrition setting and found almost 90% met the US standards required of commercial formula. Only one sample exceeded 10(5) and NONE exceeded 10(6). These were families prepared BTF using expected safe food handling practices in the US in their homes.

  1. The Madden paper indicates PCR technique identified several bacterial species but the only counts reported were Bacillus cereus (which was in acceptable limit) and Enterobacteriaceae of approximately half the blended feeds was categorised as unsatisfactory, i.e. CFU/g >10(4) but for no apparent reason (possibly a particular recipe item was contaminated at baseline).

  1. Jalili (ref 20) 98% of the BTF was contaminated at baseline (indicates unsafe food handling practice or a contaminated BTF ingredient).

  1.  Finally, Arias compared tube feeding substrates from 5 different hospitals in Costa Rica in 2003.  He found unacceptably high levels of contamination throughout the samples and importantly: " in this study no differences in the levels of contamination of the formulas were observed. Complex preparations such as those prepared with fruits or cooked vegetables showed contamination levels as high as the ones present in the commercial based solutions (Ensure® )"  Furthermore, he stated since contaminating microbes were subject to destruction by heat, the contamination had to be from improper handling/prep methods instead of the feeding substrate.

Response: Thank you for your comments. Regarding the issues raised in relation to the different studies highlighted above, we have created Table 3 which summarises the results of the different studies with respect to the level of microbial contamination of blenderised ENF compared with commercial ENF and/or international standards. This table has captured the findings of the different studies as reported by the primary researchers.

We are completely independent researchers and have only reviewed and reported the findings of articles as reported by the authors who conducted the primary research.

Comments: Overall, I feel that the paper is unjustifiably slanted against BTF use.  The studies over all show that if safe food handling is used, BTF can be safe.  It also shows that if safe food handling methods are not used, commercial formulas can be unsafe.  There is a growing body of work demonstrating the safety and efficacy of whole food based tube feeding due to its positive impact on the gut microbiome.  This review represents a wonderful opportunity to showcase the fact that BTF can be given and the benefits enjoyed IF safe food handling is employed. Patient environments and patient capacity to provide safe tube feeding need to be assessed and oversight by trained dietitians is vital for all tube feeding. It also should point out that just because commercial formula is used, it doesn't mean the patients are getting a safe feeding. 

Response: We are completely independent researchers and have only reviewed and reported the findings of articles as reported by the authors who conducted the primary research.

We feel we have adequately addressed the issue of safe food handling of blenderised and commercial ENF in relation to food safety in the discussion. The usefulness of blendersied ENF is also sufficiently addressed in the introduction. Please refer to our introduction and discussion.

Comments: Finally and most importantly, not one single study has been published showing a connection to bacterial/food borne-illness from patients on BTF.  If this has happened we certainly need to know about it, but none of the articles reviewed here report it. 

Response: The articles we reviewed did not report bacterial borne illness.

Comments: A survey of British dietitians found they all had concerns about bacterial infection of patients using BTF but none had ever encountered it in their patients. 

Response: The articles we reviewed did not report bacterial borne illness.

Reviewer 2 Report

Thank you for submitting this interesting paper. I have a few suggestion that I hope will improve it.

Abstract: I would not use abbreviations in the abstract. The words should be spelled out in full with abbreviations only in the full text of the manuscript.

You state that most studies (line 30) show issues but in the text only list 6/12. This is 50% and therefore not most. In addition, the phrasing of the sentences in line 30 , 31 and 32 are difficult to understand and have been cut and pasted directly from the more comprehensive text in the main document. Could you rewrite these sentences for the abstract reader eg "50% of studies showed microbial contamination in blenderised ENF and storage time was another important factor in the rates of contamination."

Introduction: There is an additional And in line 87. The final sentence of the introduction is not needed and is repetition of what is included in the methods. it could be removed.

Methods: There is significant repetition between the text and the table (1). I would consider omitting the paragraphs from lines 103-112 as they are not required.

Results: It would be helpful to have a summary table of the results, listing each study and whether there was an increase or not of contamination.

Discussion: What is missing here is the so what of the project. Does the contamination make a difference to the patient, especially given the benefits noted earlier. There needs to be some mention of this in the discussion and possibly in the limitations if not of the studies to date have looked at this. it is also a possible direction for future work.

Minor typos; the capital R (Review) is the title (line 4) should be changed to r. Boolean (line 115) should have a capital B.

Author Response

Reviewer 2:

Comments: Thank you for submitting this interesting paper. I have a few suggestion that I hope will improve it.

Response: Thank you for your comments.

Comments: Abstract: I would not use abbreviations in the abstract. The words should be spelled out in full with abbreviations only in the full text of the manuscript.

Response: Based on your suggestion and as appropariate, abbreviations have not been used in the abstract. 

Comments: You state that most studies (line 30) show issues but in the text only list 6/12. This is 50% and therefore not most. In addition, the phrasing of the sentences in line 30 , 31 and 32 are difficult to understand and have been cut and pasted directly from the more comprehensive text in the main document. Could you rewrite these sentences for the abstract reader eg "50% of studies showed microbial contamination in blenderised ENF and storage time was another important factor in the rates of contamination."

Response: As recommended, the sentences have been revised as follows;

“72.7% of the studies showed microbial contamination in blenderised ENF compared with 57.1% of commercial ENF and the storage time was another important factor in the rates of contamination”.

Comments: Introduction: There is an additional And in line 87. The final sentence of the introduction is not needed and is repetition of what is included in the methods. it could be removed.

Response: The ‘and’ in line 87 has been removed and the final sentence of the introduction (outlined below) has also been deleted as suggested;

“This approach involved comparing the microbial quality of blenderised ENF with commercial ENF, by applying international guidelines on microbial quality of enteral feeds as the standard, and the effect of storage time on microbial quality of blenderised ENF”.

Comments: Methods: There is significant repetition between the text and the table (1). I would consider omitting the paragraphs from lines 103-112 as they are not required.

Response: Thank you for your suggestion. We have removed the following from the method section;

Secondary Outcomes: Other Adverse Events – diarrhoea, constipation, bloating, vomiting

However, we believe that the other paragraphs are important in order to maintain the clarity of our review and we have employed the usual approach to systematic review.

Comments: Results: It would be helpful to have a summary table of the results, listing each study and whether there was an increase or not of contamination.

Response: As suggested, a summary table of the results has been developed (Please refer to table 3 in the manuscript)

Table 3: Summary of results of the level of microbial contamination of blenderised ENF compared with commercial ENF and/or international standards.

Citation

Level of Microbial Contamination of Blenderised ENF

Level of Microbial Contamination of Commercial ENF

Comments

Arias et al [33]

The concentration of Gram-negative rods found in the samples of enteral feeding solutions ranged from 103 to 107 CFU/mL, significantly exceeding the permissible level (102 CFU/mL or less). There were no differences in the levels of contamination of the formulas. Blenderised ENF made from fruits or cooked vegetables showed contamination levels as high as the ones present in the commercial based solutions (Ensure®).

Baniardalan et al [34]

The contamination of commercial formulas in all three microbiological samples was significantly more than that for handmade samples. 76% of handmade samples had total viable counts greater than 103 CFU/g compared to 96% of commercial formulas at the time of preparation.

Jalali et al [20]

Not Applicable

In the standard plate count, 97% of the samples had counts greater than 103 CFU/g, while 71% had counts greater than 104 CFU/g at the time of preparation.

Johnson et al [25]

Total bacterial count was well below acceptable limits

Total bacterial count was well below acceptable limits

All 3 feeding formulas at zero hour, 2 hours, and 4 hours for each of the 3 sampling dates were acceptable for human consumption.

Madden et al [17]

Not Applicable

The bacterial load of Enterobacteriaceae of approximately half the blended feeds was categorised as unsatisfactory (i.e. CFU/g >104), with no clear pattern of association with preparation or storage method.

Milton et al [28]

At time of preparation and 24 hours, 10% had a CFU count of >104, and at 48 hours, 12% exceeded 104 CFUs.

Not Applicable

The result showed that 88% of the samples met the US Food Code criteria for safe food consumption; 10.7% met guidelines for marginal safety by other standards, and 1.3% slightly exceeded 105 CFUs.

Mahinkazemi et al [29]

Bacterial contamination (S. aureus, coliform) of blenderised ENF at the time of preparation and 18 hrs after preparation were <101 CFU and <2MPN/g respectively.

33% of commercial ENF which were prepared in the wards had coliform contamination 6.41±2.43 MPN/g and E. coli was detected.

The presence of E. coli and coliforms in 33.3% of commercial ENF showed that these were at an unacceptable level of contamination.

Mokhalalati et al [35]

86% of standard blenderised ENF and therapeutic blenderised ENF had Aerobic Plate Count (APC) >104 CFU/g, while for all commercial ENFs were not detectable (<10 CFU/g) at all times. The maximum coliform count for any blenderised ENF sample from sites 1 and 2 was 50 CFU/g. Coliform counts for all Commercial ENFs were non-detectable (<10 CFU/g).

Pinto et al [36]

Both kinds of blenderised ENF and commercial ENF showed contamination by coliforms and Pseudomonas spp. Listeria spp. was detected in only one sample of handmade diets. However, contamination was significantly higher in the blenderised ENF (p < 0.05) compared with Commercial ENF.

Sullivan et al [37]

Not Applicable

38% of blenderised ENF had coliform counts greater than 10 MPN/g, and 92% of the samples had standard plate counts greater than 103 CFU/g. There were significant increases in mean coliform and standard plate counts over 4hours.

Vieira et al [38]

Samples of powder commercial ENF complied less when compared to liquid commercial ENF.

Only 6% of samples of blenderised ENF met the standard for coliform.

Abbreviations/Symbols: Colony-forming units (CFU); Enteral nutrition formula (ENF); Most probable number (MPN); High level of contamination (↑); Low or non-detectable level of contamination (↓).

Comments: Discussion: What is missing here is the so what of the project. Does the contamination make a difference to the patient, especially given the benefits noted earlier. There needs to be some mention of this in the discussion and possibly in the limitations if not of the studies to date have looked at this. it is also a possible direction for future work.

Response: We have included as part of our recommendations the following sentence;

Furthermore, it will be useful for future studies to explore the effect of poor handling of ENF on health outcomes.

Comments: Minor typos; the capital R (Review) is the title (line 4) should be changed to r. Boolean (line 115) should have a capital B.

Response: Thank you. The minor typos have been corrected.

Round 2

Reviewer 1 Report

Article is much improved and a welcome addition to the literature on blenderized tube feeding.

Minor suggestions:

  1. The 2015 Hurt article has been appropriately removed from the lit review table but is still included in Appendix 1.  It should be removed from Appendix 1.
  2. Table 2 has an abbreviation explanation for Head and Neck cancer but this term doesn't appear in the table.

Author Response

Thank you for these final comments, we can confirm the abbreviation from Table 2, which did not appear in the table has been removed, and Hunt et al. paper which has been removed from the review has now been removed from appendix 1.

Thank you once again,

Round 3

Reviewer 1 Report

Thank you for the opportunity to review this article.

This manuscript is a resubmission of an earlier submission. The following is a list of the peer review reports and author responses from that submission.

Round 1

Reviewer 1 Report

Thank you for the opportunity to review this article.  I will not comment on the statistical methodology as some of the terms are not familiar to me (example statistic I(2).

The authors have reached the conventional conclusion that blenderized tube feeding (BTF) is not sanitary though they agree the literature is mixed. Indeed, the overall p-value was 0.04.  They conclude that ready to hang commercial formulas are the most appropriate enteral feeding. However, perhaps the conclusion should be that BTF preparation techniques need to be established and caregivers taught how to prepare and administer it appropriately in order to reduce contamination. Reports of improved tolerance to BTF compared to commercial standard formulas are widely reported. It would not be ethical to deny tube fed patients the advantages of a BTF that is prepared correctly.

The authors should explain the increased interest and use of BTF is due to the fact that caregivers and patients report better tolerance to a real food blend (reduced gagging, retching, vomiting, diarrhea, constipation) and overall improved achievement of weight/growth goals. Line 88 on page 2 states "the potential benefits of blenderised ENF have not been well researched" but that is not true.  There are several convincing publications demonstrating the superiority of a whole food blend over commercial formula (see ref list below).

Nutrition experts tout the advantages of a whole food/plant based diet that confers intestinal microbiome diversity and improved health outcomes; in contrast, traditional commercial formula is highly processed and monotonous with no representation from plant sources. The message that a whole food/plant based diet is advantageous has reached the tube feeding population and they want to utilize this feeding substrate. Accordingly, the enteral formula industry has responded with many commercially-prepared whole food blends that are available in a ready to hang form (Bennett K, Hjelmgren B, Piazza J. Blenderized tube feeding: health outcomes and review of homemade and commercially prepared products. Nutr Clin Prac. 2020;35(3):417-431).  Furthermore, a BTF can be made by adding sterile baby food to a commercial formula- providing advantages of food and advantages of sterile products.

Note that prior to the work by Johnson and the work by Milton, BTF microbial studies were conducted in countries and conditions where the ambient temperatures are higher than expected in hospitals and homes of developed countries. Also, the methods of BTF preparation in those studies were not described so we cannot be sure that they were prepared properly. More importantly, no study has specifically linked BTF to increased food borne illness in tube fed patients. Another key feature of these studies is that subjects receiving commercial formula also had more oversight from healthcare providers because they had better insurance/medical provider programs.  Many of the patients using BTF did not.  Perhaps it is oversight and guidance on enteral feeding that makes the difference instead of the feeding substrate.

Finally, I am not sure why the 2015 Hurt study is included in the analysis. There was no investigation of bacterial load of BTF. This was a survey of their patients on home enteral nutrition- many using full or partial blended tube feeding.  No formula was collected and tested. The survey showed that participants experienced much better outcomes on BTF compared to standard commercial enteral formula (see abstract below).

Overall I think this work is valuable but should include the rationale for using BTF, reach a different conclusion, and the Hurt study eliminated from the meta-analysis.

  1. Coad J, Toft A, Lapwood S, Manning J, Hunter M, Jenkins H,... Widdas, D. Blended foods for tube-fed children: A safe and realistic option? A rapid review of the evidence. Arch Dis Child. 2017;102:274-278.
  2. Hron B, Fishman E, Lurie M, et al. Health Outcomes and Quality of Life Indices of Children Receiving Blenderized Feeds via Enteral Tube. J Pediatr. 2019;211:139-145.
  3. Carter H, Johnson K, Johnson TW, Spurlock A. Blended tube feeding prevalence, efficacy, and safety: What does the literature say? J Am Assoc Nur Pract. 2018;30(3)3:150–157.
  4. Kariya C, Bell K, Bellamy C, Lau J, Yee K. Blenderized Tube Feeding: A Survey of Dietitians' Perspectives, Education, and Perceived Competence. Can J Diet Pract Res. 2019;80(4):190-194.
  5. Johnson T, Spurlock A, Epp L, Hurt RT, Mundi MS. Reemergence of Blended Tube Feeding and Parent's Reported Experiences in Their Tube Fed Children. J Alt Com Med. 2018;24(4):369-373.
  6. Epp L, Lammert L, Vallumsetla N, Hurt RT, Mundi M. Use of blenderized tube feeding in adult and pediatric home enteral nutrition patients. Nutr Clin Prac. 2017;32(2):201-205.
  7. Hurt RT, Edakkanambeth VJ, Epp L, Pattinson AK, Lammert LM,  Lintz, JE, Mundi MS. Blenderized tube feeding use in adult home enteral nutrition patients: A cross-sectional study. Nutr Clin Prac. 2015;30(6):824-829.
  8. Armstrong J, Buchanan E, Duncan H, Ross K, Gerasimidis K. Dietitians' perceptions and experience of blenderised feeds for paediatric tube-feeding. Arch Dis Child. 2016;102:152-156.
  9. Johnson TW, Spurlock A, Pierce L. Survey study assessing attitudes and experiences of pediatric registered dietitians regarding blenderized food by gastrostomy tube feeding. Nutr Clin Prac. 2015;30(3):402-405.
  10. Samela K, Mokha J, Emerick K, Davidovics ZH. Transition to a tube feeding formula with real food ingredients in pediatric patients with intestinal failure. Nutr Clin Prac. 2016;32(2):277-281.
  11. Gallagher K, Mouzaki M, Carpenter A, Haliburton B, Bannister L, Norgrove H, Hoffman L, Maroon M. The BLEND Study: A feasibility study looking at children transitioning to blenderized tube feeds. [Supplemental Material]. J Ped Gastr Nutr. 2015;61:206-207.
  12. Klek S, Hermanowicz A, Dziwiszek G, Matysiak K, Szczepanek K., Szybinski P, Galas A. Home enteral nutrition reduces complications, length of stay, and health care costs: Results from a multicenter study. Am J Clin 2014; 100(2):609-615.
  13. Pentiuk S, O’Flaherty T, Santoro K, Willging, P, Kaul A. Pureed by gastrostomy tube diet improves gagging and retching in children with fundoplication. J Par Ent Nutr. 2011; 35(3):375-379.
  14. Batsis I, Davis, L, Prichett L, Wu L, Shores D, Yeung K, Oliva-Hemker M. Efficacy and tolerance of blended diets in children receiving gastrostomy feeds. Nutr Clin Prac. 2020;35(2):282-288.
  15. Bennett K, Hjelmgren B, Piazza J. Blenderized tube feeding: health outcomes and review of homemade and commercially prepared products. Nutr Clin Prac. 2020;35(3):417-431.
  16. Gallagher K, Flint A, Mouzaki M, Carpenter A, Haliburton B, Bannister L, Norgrove H, Hoffman L, Mack D, Stintzi A, Marcon M. Blenderized Enteral Nutrition Diet Study: Feasibility, Clinical, and Microbiome Outcomes of Providing Blenderized Feeds Through a Gastric Tube in a Medically Complex Pediatric Population. J Parenter Enteral Nutr. 2018 Aug;42(6):1046-1060.
  17.  

Ryan T. Hurt, MD, PhD1,2,3,4; Jithinraj Edakkanambeth Varayil, MD1,5;
Lisa M. Epp, RD3; Adele K. Pattinson, RD3; Lisa M. Lammert, RD3;
Jennifer E. Lintz, RD3; and Manpreet S. Mundi, MD3
Abstract
Background:Use of long-term enteral nutrition (EN) has increased dramatically in the United States. It has been the authors’ experience that most home EN (HEN) patients use blenderized tube feeding (BTF) in addition to commercial EN. There are limited resources available for patients interested in BTF, and studies evaluating safety and effectiveness are limited. Methods: The authors conducted a prospective cross-sectional study (n = 54). Inclusion criteria: age >18 years, follow-up in HEN clinic, prescribed commercial EN. Participants were provided the survey at HEN follow-up appointments after receiving HEN for at least 3 weeks. Results: Median age (range) was 60.5 (22–87) years with 42.6% females (n = 23). BTF was used by 55.5% of patients (n = 30). Most (57%; n = 31) received HEN for >6 months. BTF use was a median of 4 (1–7) days per week. Most common reasons for using BTF were as follows: it is more natural (43%), like eating what their family does (33%), and tolerate BTF better (30%). In patients who use BTF, 80% reported maintaining goal body weight. BTF resulted in significantly less reported nausea, vomiting, bloating, diarrhea, and constipation compared with commercial EN. Conclusions: This is the first study to evaluate BTF use in an adult HEN population. More than 50% of our patients used and approximately 80% expressed a desire to use BTF if provided with adequate information. With new connection tube changes coming in the near future, adequate adapters for BTF need to be developed.

Again, thank you for the opportunity to review this timely work.

Reviewer 2 Report

The manuscript reports an analysis of blenderised/artisanal and commercial enteral diets concerning microbiological quality. The theme has scientific relevance.

The introduction is adequate, and the objective is clear. The methodology is appropriate and scientifically consistent. However, the results obtained are questionable due to the number of studies eligible to infer about the microbiological quality of diets. The microbial quality of  Blenderised ENF is greatly influenced by the good manufacturing practices, and I understand that to obtain consistent and scientifically relevant evidence the meta-analysis must have an eligible number of studies greater than two or three.

Regarding the objective"Microbial Quality of Blenderised ENF Versus Commercial ENF": The results were obtained in only two studies, which analyzed only 51 samples of Blenderised ENF  and 16 Commercial ENF. A small number of samples to establish evidence about the objective. Considering the reference Mokhalalati et al. 2004,  blenderised ENF diets were analyzed in only three production units (hospitals) and the commercial ENF diets in only one hospital, and most likely, these samples are from the same production lot. This quantity of analyzed samples is insufficient to validate evidence of the microbiological quality of food.

Mokhalalati et al. 2004 report that "Aerobic plate counts for all CPF samples were non detectable", i.e., it is not possible to affirm the result found (mean=11.5). Another consideration is on the estimation of the average population of bacteria, it is known that a microbial population has exponential growth and the method chosen to estimate the mean, from the media and percentiles, can significantly influence the inference of the mean, since most approaches consider that the data come from a normal distribution. I suggest the authors specify how they estimated the average microbial population.

Regarding Figure 3, I understand that the total number of samples eligible for the study by Mahinkazemi et al. 20017 is 18 and not 36 samples. Again, I would like to point out the consideration about estimating the average population of bacteria, especially for coliforms in the study by Mahinkazemi  et al. 20017, since coliforms were not detected in any of the Blenderised ENF samples analyzed. For coliforms, the result (Figure 3) shows a great heterogeneity, a considerable bias for the result of the evidence.

Reviewer 3 Report

Overall a well written manuscript covering an important topic.  I am unclear as to why pediatric data was excluded.  The title should be changed to reflect that this only included adult data.  Otherwise, I only have a few recommendations. 

Abstract

Line 21: change to “The use of blenderized…”

Line 56: re-evaluate your definition of blenderized tube feed as there are commercial BTF formulas available.  Consider revising to “which is any whole food mixture or liquid that is blenderized to be provided through enteral tube and does not consist of water, medicine, or standard commercial enteral feeds, is on the increase worldwide…

Line 59: add “at home by blending whole food or meal into a liquid”

Discussion
please also discuss whether there was a difference in the studies from developing versus developed countries.  Many of the early studies are also now from a few years ago and storage and refrigeration techniques could also have skewed the results. 

Limitations

Please also add why pediatric studies were excluded.  The intent of the meta-analysis is to evaluate the microbial quality and safety of blenderized enteral nutrition formula.  I’m not sure why pediatric studies were excluded.  This dramatically reduces generalizability.